

# Independent evolution of tetraloop in enterovirus oriL replicative element and its putative binding partners in virus protein 3C

Maria A. Prostova[1], Andrei A. Deviatkin[1], Irina O. Tcelykh[1,2], Alexander N. Lukashev[1,3] and Anatoly P. Gmyl[1,2,3]

[1] Chumakov Institute of Poliomyelitis and Viral Encephalitides, Moscow, Russia
[2] Lomonosov Moscow State University, Moscow, Russia
[3] Sechenov First Moscow State Medical University, Moscow, Russia

## ABSTRACT

**Background.** Enteroviruses are small non-enveloped viruses with a (+) ssRNA genome with one open reading frame. Enterovirus protein 3C (or 3CD for some species) binds the replicative element oriL to initiate replication. The replication of enteroviruses features a low-fidelity process, which allows the virus to adapt to the changing environment on the one hand, and requires additional mechanisms to maintain the genome stability on the other. Structural disturbances in the apical region of oriL domain d can be compensated by amino acid substitutions in positions 154 or 156 of 3C (amino acid numeration corresponds to poliovirus 3C), thus suggesting the co-evolution of these interacting sequences in nature. The aim of this work was to understand co-evolution patterns of two interacting replication machinery elements in enteroviruses, the apical region of oriL domain d and its putative binding partners in the 3C protein.

**Methods.** To evaluate the variability of the domain d loop sequence we retrieved all available full enterovirus sequences ($>6,400$ nucleotides), which were present in the NCBI database on February 2017 and analysed the variety and abundance of sequences in domain d of the replicative element oriL and in the protein 3C.

**Results.** A total of 2,842 full genome sequences was analysed. The majority of domain d apical loops were tetraloops, which belonged to consensus YNHG (Y = U/C, N = any nucleotide, H = A/C/U). The putative RNA-binding tripeptide 154–156 (*Enterovirus C* 3C protein numeration) was less diverse than the apical domain d loop region and, in contrast to it, was species-specific.

**Discussion.** Despite the suggestion that the RNA-binding tripeptide interacts with the apical region of domain d, they evolve independently in nature. Together, our data indicate the plastic evolution of both interplayers of 3C-oriL recognition.

## INTRODUCTION

Enteroviruses are small non-enveloped viruses with a plus strand genome about 7500 nt long which contains one open reading frame that encodes structural (capsid) and non-structural proteins, 5′ and 3′ NTRs (non translated regions), and polyA on the 3′ end

Corresponding author
Maria A. Prostova,
prostova_ma@chumakovs.su,
prostovna@gmail.com

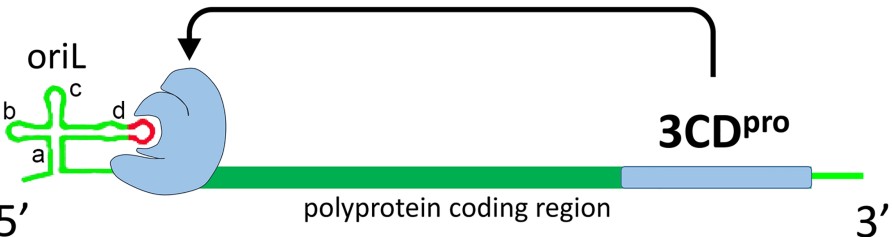

**Figure 1** Schematic representation of interaction of poliovirus protein 3CD (colored with blue) with poliovirus genome replicative element oriL. Domains a, b, c and d of oriL are labeled. Apical region of domain d, corresponding to the tetraloop and its flanking base pair, is colored with red.

(*Palmenberg, Neubauer & Skern, 2010*) (Fig. 1). Most non-structural enterovirus proteins are polyfunctional. Protease 3CD is a precursor of polymerase 3D and plays a key role in the initiation of replication (*Harris et al., 1992*; *Gamarnik & Andino, 1998*; *Thompson & Peersen, 2004*). After translation by host cell ribosomal machinery, the genome is utilized for the synthesis of the (−) strand RNA, which, in turn, serves as a matrix for the synthesis of multiple daughter (+) strands. Non-translated regions of the genome and a coding sequence within the genomic region encoding the viral helicase 2C contain replicative elements, which interact with viral and host proteins. These RNA-protein complexes regulate initiation and further steps of replication. For poliovirus, the most clinically relevant member of the *Enterovirus* genus, there are at least three known RNA-protein complexes, which are formed with the replicative elements oriL, oriR and oriI during replication (Fig. 1).

Complex oriL with viral protein 3CD and the host protein PCBP2 is crucial for the transcription initiation (*Goodfellow et al., 2000*; *Vogt & Andino, 2010*; *Chase, Daijogo & Semler, 2014*). The element oriL has a cloverleaf-like secondary structure with four domains termed a (stem of the cloverleaf), b, c and d (leaves of the cloverleaf) (*Trono, Andino & Baltimore, 1988*; *Andino, Rieckhof & Baltimore, 1990*) (Fig. 1). Previously, it was demonstrated *in vitro* that 3CD (or 3C) of poliovirus, coxsackievirus B3 and bovine enterovirus 1 interacts with the apical loop and the flanking base pairs of hairpin d in the oriL element (*Andino, Rieckhof & Baltimore, 1990*; *Du et al., 2003*; *Ihle et al., 2005*) (Fig. 1).

The apical loops of domain d in genomes belonging to several viruses of *Enterovirus* genera were shown by NMR experiments to be tetraloops with a specific spatial structure, which belongs to the UNCG structural class of stable tetraloops (*Du et al., 2003*; *Du et al., 2004*; *Ihle et al., 2005*; *Melchers et al., 2006*). There are several known structural classes of tetraloops, three of which, named according to consensus sequences, contain tetraloops of extreme stability: UNCG (where N = any nucleotide), GNRA (where R = A/G) and gCUUGc (*Uhlenbeck, 1990*; *Cheong & Cheong, 2010*). Tetraloops of UNCG and GNRA classes are the most widely represented (*Woese et al., 1990*; *Cheong & Cheong, 2010*; *Bottaro & Lindorff-Larsen, 2017*). Previously, it was shown that only tetraloops of the UNCG structural class, but not tetraloops of GNRA or gCUUGc structural classes, can support effective replication of the poliovirus genome (*Prostova et al., 2015*). Moreover, the exact sequence of the apical region of poliovirus domain d was of less importance for effective

3CD-oriL recognition than its spatial structure (*Rieder et al., 2003*; *Prostova et al., 2015*). At the same time, structural disturbance in the apical region of oriL domain d of poliovirus could be compensated by amino acid substitutions in the tripeptide 154–156 of the 3C protein (here and hereafter amino acid numeration corresponds to poliovirus 3C protein) (*Andino et al., 1990*; *Prostova et al., 2015*). In addition to triplet 154–156, the conserved motif $_{82}$KFRDI$_{86}$ of the 3C protein also takes part in the oriL recognition (*Andino et al., 1990*; *Andino et al., 1993*; *Hämmerle, Molla & Wimmer, 1992*; *Shih, Chen & Wu, 2004*). To date, a comprehensive analysis of the diversity in domain d apical region and amino acid tripeptide sequences in the *Enterovirus* genus has not been conducted.

The replication of an enterovirus is a low-fidelity process, generating, on average, one mutation per genome (*Sanjuán et al., 2010*; *Acevedo, Brodsky & Andino, 2013*). The high probability of mutation allows the virus to adapt to a constantly changing environment on the one hand, but requires additional mechanisms to maintain genome stability on the other (*Wagner & Stadler, 1999*; *Lauring, Frydman & Andino, 2013*). The aim of the present study was to understand co-evolution patterns of two interacting replication machinery elements in enteroviruses, the apical domain d of oriL and the 3C protein.

## MATERIALS AND METHODS

### Formation and filtration of sets of full genomes

All available nucleotide sequences (as of February 2017) containing the *Enterovirus* genus with length 8000>n>6800 were extracted from the NCBI database. For every species, a multiple sequence alignment was conducted using MAFFT version 7 with default settings (*Katoh & Standley, 2013*). Sequences that contained more than 50 N characters in succession and sequences that were annotated as "Modified_Microbial_Nucleic_Acid", were removed from alignments. All sequences that differed from any other sequence in the dataset by less than 1% of the nucleotide sequence were omitted in order to reduce the bias caused by over-represented sequences.

### Analysis of tetraloop and amino acid variety in the sets of genomes

For analysis of domain d sequence variety, the multiple sequence alignments were used. The relevant region of multiple sequence alignment and the respective names of sequences were analysed in Microsoft Excel. To analyse correlation of the domain d loop and tripeptide of 3C sequences the same alignments were translated in the protein 3C coding region. The resulting amino acid sequences that corresponded to tripeptides 154–156 (poliovirus 3C numerations) were analysed using Microsoft Excel. An amino acid frequency plot was created via the WebLogo server using the set of filtered genomes for every species (*Crooks et al., 2004*). To do this, the multiple sequence alignment of filtered genomes of every species was translated in the region that codes protein 3C, while positions 71–89 and 147–160 were saved in separate MAS files, which were then used to produce logos.

### Domain d secondary structure

The domain d secondary structure was folded using the Vienna RNA Websuite server with subsequent manual editing (*Gruber et al., 2008*; *Lorenz et al., 2011*). Algorithm accounting for minimum free energy and partition function was used.

**Table 1  Number of full genome sequences that contained oriL region and number of unique domain d sequences before and after filtration.** For *Enterovirus E* and *F* number of unique tetraloops is shown separately for first and the second oriL.

| Species | Number of full genome sequences | Number of full genome sequences after 1% nucleic identity filtration | Number of unique tetraloops before filtration | | Number of unique tetraloops after filtration | |
|---|---|---|---|---|---|---|
| *Enterovirus A* | 1052 | 564 | | 17 | | 16 |
| *Enterovirus B* | 339 | 244 | | 18 | | 18 |
| *Enterovirus C* | 747 | 274 | | 15 | | 12 |
| *Enterovirus D* | 419 | 57 | | 7 | | 6 |
| *Enterovirus E* | 12 | 10 | 6 | 5 | 6 | 5 |
| *Enterovirus F* | 13 | 10 | 4 | 3 | 4 | 3 |
| *Enterovirus G* | 10 | 8 | | 6 | | 6 |
| *Enterovirus H* | 3 | 2 | | 2 | | 2 |
| *Enterovirus J* | 8 | 5 | | 3 | | 3 |
| *Rhinovirus A* | 159 | 118 | | 8 | | 8 |
| *Rhinovirus B* | 50 | 37 | | 7 | | 7 |
| *Rhinovirus C* | 38 | 37 | | 6 | | 6 |

# RESULTS

## Sample characteristics

To evaluate the variability of the domain d loop sequence we retrieved all available complete genome ($8000>n>6800$ nucleotides) enterovirus (EV) sequences that were present in the NCBI database on February 2017. Representatives of *Enterovirus A* (1173 sequences in total), *Enterovirus B* (414), *Enterovirus C* (773), *Enterovirus D* (462), *Enterovirus E* (12), *Enterovirus F* (13), *Enterovirus G* (15), *Enterovirus H* (3), *Enterovirus J* (7), *Rhinovirus A* (202), *Rhinovirus B* (76) and *Rhinovirus C* (51) species were analysed. As expected, genomes of epidemiologically significant viruses were the most represented in the database. For example, 66% of *Enterovirus A* species genomes belonged to the EV71 type, the causative agent of hand, foot and mouth disease (*Solomon et al., 2010*); most *Enterovirus C* species sequences (78%) belonged to poliovirus; and most *Enterovirus D* species sequences (98.7%) represented EV68, an aetiological agent of severe respiratory illness (*Oermann et al., 2015*). The number of genome sequences of each species that contained the oriL region is shown in Table 1.

Sequences of apical regions in oriL domain d and the amino acids involved in RNA recognition in 3C protein were analysed. In genomes of *Enterovirus E* and *Enterovirus F* species with two oriLs (*Pilipenko, Blinov & Agol, 1990*; *Zell et al., 1999*) sequences of both oriLs were analysed (Table 1). To reduce the bias towards particular loop sequences present in a large set of closely related genomes, which, for example, belonged to one outbreak, all sequences that differed from any other sequence in the dataset by less than 1% of the nucleotide sequence were omitted. After curation, the sizes of the largest data sets decreased dramatically, but the number of unique loop sequences in every set did not change significantly (Table 1). Unique tetraloop variants were lost for *Enterovirus A*

Peer J

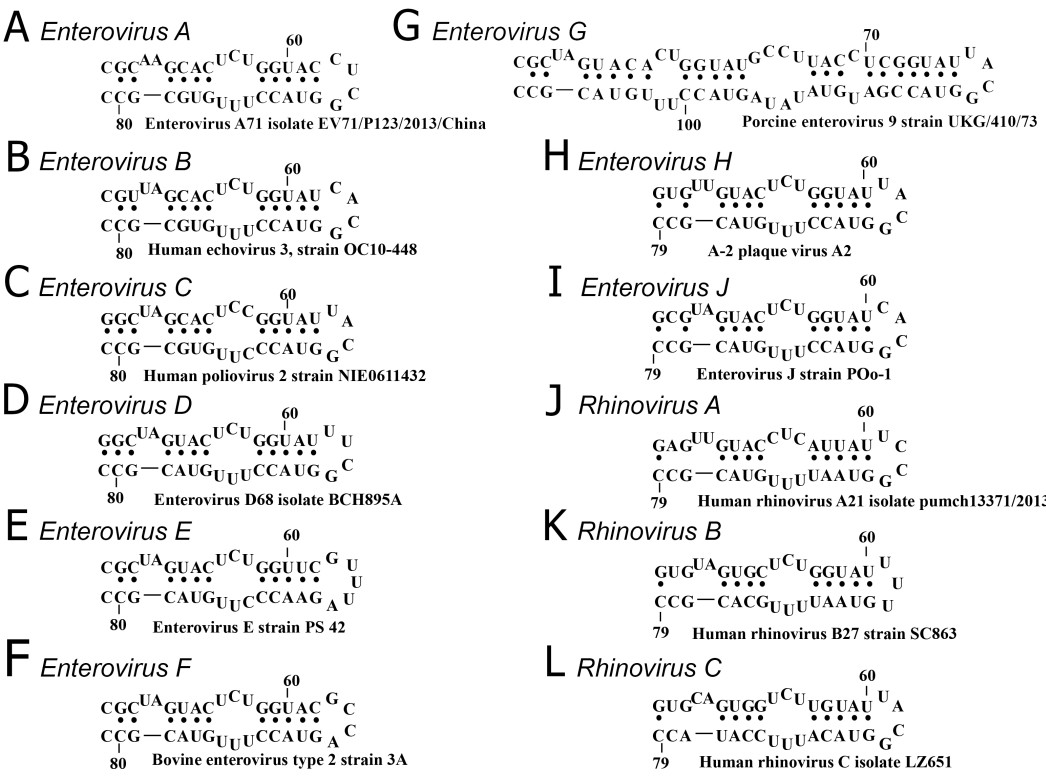

**Figure 2** **Secondary structure of oriL domain d of distinct enterovirus species.** (A–I) secondary structure of oriL domain d in *Enterovirus A-J* genome. For *Enterovirus E* and *F* domain d of the first oriL is shown. Secondary structure of domain d of Porcine enterovirus 9 strain UKG/410/73 was folded with use as reference (*Krumbholz et al., 2002*); (J–L) secondary structure of oriL domain d in *Rhinovirus A-C* genome.

(tetraloop UGUG), *Enterovirus C* (tetraloops CCCG, CAUG and UGUG) and *Enterovirus D* (tetraloop UUGG). This indicates that, even among closely related genomes, the tetraloop sequence can vary. Indeed, in several outbreaks caused by EV71 or PV1, closely related genomes contained different apical domain d sequences (not shown). It should be noted that the filtration of the dataset using a 95% sequence identity threshold resulted in a dramatic loss of unique tetraloop variants (107 genomes out of 1,052 were left after filtration, while 13 unique tetraloop variants out of total 17 variants were detected in the filtered set).

## Variability in the oriL domain d apical loop sequence

The secondary structure of domain d was conserved in all species, except *Enterovirus G*, in which an elongated domain d was observed (Fig. 2) (*Krumbholz et al., 2002*).

The variety and occurrence of various loops in the apical region of domain d in all species of the *Enterovirus* genera were analysed in filtered sets of full genome sequences. Most domain d apical loops were tetraloops (i.e., they consisted of four nucleotides) (Table 2). However, triloops (3-nucleotide loop) could be found in genomes of *Enterovirus C* and *Rhinovirus A* and *B* species, whereas pentaloops (5-nucleotide loop) were detected in genomes of Enterovirus E species (Table 2).

**Table 2 Occurrence of domain d apical sequences in filtered sets of full genomes of different enterovirus species and serotypes.** Tetraloops CCCG, UGUG, CAUG and UUGG that were unique for species *Enterovirus A, B, C* and *D* and were lost upon filtration, were added to maintain the diversity of loop sequence and are shown in blue. The gradient coloring from red to green represents abundance heat map for the genomes with different domain d sequence.

| Loop sequence | Enterovirus | | | | | | | | | | | | | | Rhinovirus | | |
|---|---|---|---|---|---|---|---|---|---|---|---|---|---|---|---|---|---|
| | A | | | | B | C | | | D | E | F | G | H | J | A | B | C |
| | all | EV71 | EV71 C4 genotype | non EV71 | | all | PV | non PV | | | | | | | | | |
| Triloops | | | | | | | | | | | | | | | | | |
| CCG | | | | | | 1 | | 1 | | | | | | | | | |
| CAG | | | | | | 1 | | 1 | | | | | | | | | |
| UCU | | | | | | | | | | | | | | | 1 | 5 | |
| UUU | | | | | | | | | | | | | | | | 17 | |
| UAU | | | | | | | | | | | | | | | | 8 | |
| AUU | | | | | | | | | | | | | | | | 4 | |
| UGU | | | | | | | | | | | | | | | | 1 | |
| UUC | | | | | | | | | | | | | | | | 1 | |
| GAU | | | | | | | | | | | | | | | | 1 | |
| YNMG Tetraloops | | | | | | | | | | | | | | | | | |
| UACG | 85 | 28 | | 57 | 51 | 106 | 64 | 42 | | | | 3 | 1 | 2 | 38 | | 15 |
| UGCG | 114 | 2 | | 112 | 31 | 43 | 31 | 12 | | | | 1 | 1 | | 2 | | |
| UUCG | 16 | 16 | 14 | | 3 | | | | 50 | | | | | | 6 | | 6 |
| UCCG | 2 | | | 2 | 11 | 1 | 1 | | | | | | | | 53 | | 10 |
| CACG | 48 | 28 | | 20 | 98 | 101 | 54 | 47 | 1 | | | 1 | | 2 | 5 | | |
| CGCG | 3 | 2 | | 1 | 3 | 13 | 6 | 7 | | | | 1 | | | | | |
| CUCG | 132 | 127 | 126 | 5 | 5 | 2 | | 2 | 2 | | | | | | 1 | | 3 |
| CCCG | 40 | 39 | 28 | 1 | 16 | 1 | | 1 | 1 | | | | | | 12 | | 2 |
| UAAG | 10 | 10 | | | 2 | | | | | | | | | | | | |
| UGAG | 22 | 22 | | | 1 | | | | | | | | | | | | |
| UUAG | | | | | | | | | | | | | | | | | |
| UCAG | | | | | | | | | | | | | | | | | |
| CAAG | 1 | 1 | | | 4 | 1 | | 1 | | | | | | 1 | | | |
| CGAG | 1 | | | 1 | | 2 | | 2 | | | | | | | | | |
| CUAG | | | | | | | | | | | | | | | | | |
| CCAG | | | | | | | | | | | | | | | | | |
| YACG | | | | | 1 | | | | | | | | | | | | |
| YNUG Tetraloops | | | | | | | | | | | | | | | | | |
| UAUG | 54 | 1 | | 53 | | | | | | | | | | | 1 | | |
| UGUG | 1 | | | 1 | 1 | 1 | | 1 | | | | | | | | | |
| UUUG | | | | | | | | | 1 | | | | | | | | |
| UCUG | | | | | 1 | | | | | | | | | | | | |
| CAUG | | | | | 9 | 1 | | 1 | | | | | | | | | |
| CGUG | 1 | | | 1 | 3 | 2 | | 2 | | | | | | | | | |
| CUUG | 34 | 34 | 35 | | 3 | | | | 2 | | | | | | | | |
| CCUG | 1 | 1 | 1 | | 1 | 1 | | 1 | | | | | | | | | |
| GYYA Tetraloops | | | | | | | | | | | | | | | | | |
| GCUA | | | | | | | | | | 2 | 13 | | | | | | |
| GCCA | | | | | | | | | | | 3 | | | | | | |
| GUUA | | | | | | | | | | 2 | 3 | 1 | | | | | |
| Other tetraloops | | | | | | | | | | | | | | | | | |
| UUGG | | | | | | | | | 1 | | | | | | | | |
| CUUC | | | | | | | | | | | | | | | | | 1 |
| AUUA | | | | | | | | | | | 1 | | | | | | |
| Pentaloops | | | | | | | | | | | | | | | | | |
| GCUUA | | | | | | | | | | 7 | | | | | | | |
| GUUUA | | | | | | | | | | 2 | | | | | | | |
| GCCUA | | | | | | | | | | 4 | | | | | | | |
| GCGUA | | | | | | | | | | 1 | | | | | | | |
| GCGUA | | | | | | | | | | 1 | | | | | | | |
| GAUUA | | | | | | | | | | 1 | | | | | | | |
| GUCUA | | | | | | | | | | 1 | | | | | | | |

| |
|---|
| 1 |
| 2 |
| 11 |
| 31 |
| 51 |
| 132 |

The most common loop sequences belonged to consensuses YNMG (Y = C/U, N = any, M = A/C; tetraloops with UNCG class spatial structure belong to this consensus) and YNUG (tetraloops with UNCG class and gCUUGc class spatial structures belong to this consensus) (Table 2). Consensus YNMG and consensus YNUG together corresponded to 24 unique sequence variants. Interestingly, in our dataset of 2,842 full genomes, four tetraloops out of 24 possible variants were never found in the domain d apical region:

UUAG, UCAG, CUAG and CCAG (Table 2). Thus, dinucleotides UA and CA are likely to be avoided at positions 2 and 3 of the tetraloop in enterovirus genomes.

In *Enterovirus A* species, 17 out of 24 possible unique tetraloop sequences were identified (Table 2, Table S2). Twelve unique loops of *Enterovirus A* belonged to consensus YNMG, while the other five belonged to consensus YNUG. The most abundant tetraloop in Enterovirus A genomes, in contrast to other species, was CUCG (Table 2, Table S1). This is explained by the prevalence of this tetraloop in the EV71 C4 genotype (Table 2, Table S1). The frequency of other tetraloop sequences varied significantly (Table 2, Table S2). One tetraloop (UGUG) was lost upon filtration. Such sequences here and below were manually added to the final data set to maintain diversity of loop sequences within the species, as well as provide comprehensive information about sequences in apical domain d in viable viruses (Table 2). Interestingly, EV71 sequences contained 13 out of 17 tetraloop variants, which were detected in the *Enterovirus A* genus (Table 2). In other words, the diversity of tetraloops in one discrete lineage in general resembles its diversity in the unification of different discrete lineages.

In Enterovirus B genomes, 18 unique tetraloops out of 24 possible were found. Twelve of these tetraloops belonged to consensus YNMG and six to consensus YNUG (Table 2, Table S3). The most abundant tetraloops were CACG (98 genomes), UACG (51 genomes) and UGCG (31 genomes), which were also present among the most abundant tetraloops of Enterovirus A species.

In genomes of the *Enterovirus C* species, nine unique tetraloops belonged to the YNMG consensus and four to the YNUG consensus. Three unique tetraloops were lost upon filtration and added to the final data set (CCCG, UGUG, CAUG) (Table 2, Table S4). Two genomes annotated in the NCBI data base as Human coxsackievirus A21, strain Coe, (accession number D00538) and Human coxsackievirus A21, strain BAN00-10467, (accession number EF015031) contained triloops CAG and CCG, respectively. The most abundant tetraloops in EV-C species were UACG (106), CACG (101), UGCG (43) (Table 2, Tables S1 and S4), which corresponds to the Sabin vaccine strains of poliovirus serotypes 2, 3 and 1, respectively. To evaluate bias caused by the redundant number of vaccine strain sequences in the data set, we subtracted genomes of vaccine/vaccine derived poliovirus strains from the analysed set. As a result, tetraloops UACG, CACG and UGCG were still the most frequent variants (Table 2, Table S1).

Only 57 Enterovirus D genomes out of 419 were left after 1% identity filtration. Fifty genomes belonged to Human enterovirus 68, the aetiological agent of respiratory illness. All genomes of this type contained loop UUCG in the domain d apical region. Other tetraloops were UUUG (1), CUCG (2), CCCG (1), CUUG (2) and CACG (1) (Table 2, Table S5). One tetraloop (UUGG) was lost upon filtration and manually added to the final data set.

Species *Enterovirus E* and *F* have two oriLs in the 5′ NTR, generally with similar sequences in the apical region of domain d (*Pilipenko, Blinov & Agol, 1990*; *Zell et al., 1999*) (Table S6). As such, we united sequences from the first and the second oriL of these viruses in the heat map (Table 2). Domain d loops in 10 genomes of Enterovirus F were tetraloops, while, in 10 Enterovirus E genomes, there were both tetraloops (first oriL) and pentaloops

(first and second oriL) (Table 2, Table S6). There were four diverse tetraloop sequences in oriLs of *Enterovirus E* and *F* with no obvious preference between these species. These sequences were GCUA, GUUA, GCCA, AUUA (Table 2, Table S6). Tetraloop AUUA was found once in the first oriL domain d of EV-F (strain PS87/Belfast, accession number DQ092794) (Table 2, Table S6). There were six diverse pentaloop sequences in domain d of Enterovirus E genomes—GCUUA, GUUUA, GCCUA, GCGUA, GAUUA, GUCUA (Table 2, Table S6).

All domain d loops in genomes of *Enterovirus G, H* and *J* species were tetraloops; all except one tetraloop variant belonged to consensus YNMG (Table 2, Table S7). One Enterovirus G representative had a GUUA tetraloop sequence (strain LP 54, accession number AF363455), similar to loops of *Enterovirus E* and *F* species (Table 2). This genome had only one oriL with the same domain d length as that of Enterovirus G genomes (*Krumbholz et al., 2002*).

All except one (isolate V38_URT-6.3m, accession number JF285329) of the full genomes of Rhinovirus A species and all full genomes of Rhinovirus C species had tetraloops in the apical regions of domain d (Table 2). Tetraloops of these viruses in almost all cases belonged to consensus YNMG, with one exception found in Rhinovirus C (tetraloop CUUC, isolate JAL-1, accession number JX291115) (Table 2, Table S8). All loops in the apical region of Rhinovirus B domain d were triloops (Table 2, Table S8).

Thus, the secondary structure of domain d was very similar among species of the *Enterovirus* genus, with the exception of *Enterovirus G* species (Fig. 2). The apical region of domain d has a high diversity of sequences; however, in species of *Enterovirus A, B, C, D, G, H* and *J* and *Rhinovirus A* and *C*, it mostly corresponds to the same consensus, that is, YNHG (Y = C/U, N = any, H = A/C/U).

## Variety of RNA-recognition tripeptide of 3C

Two motifs of protein 3C are involved in RNA recognition and interact with oriL: the conservative motif KFRDI (positions 82–86 of poliovirus 3C) and the putative RNA-binding tripeptide (positions 154–156 in poliovirus 3C) (*Andino et al., 1990*; *Andino et al., 1993*; *Hämmerle, Hellen & Wimmer, 1991*; *Shih, Chen & Wu, 2004*). Substitutions in the putative RNA-binding tripeptide are known to compensate for disturbance in the apical region of domain d, such that the RNA-binding tripeptide is a putative candidate to co-evolve with the domain d loop (*Andino et al., 1990*; *Prostova et al., 2015*). There are other amino acids that have been found to affect oriL-3CD interaction, but tripeptide 154–156 (*Enterovirus C* 3C protein numbering here and below) is the only one that is proven to compensate structural disturbance in the domain d apical region (*Andino et al., 1990*; *Andino et al., 1993*). To evaluate the possible co-evolution between the domain d tetraloop and its putative interaction partners in protein 3C, relevant sequences in the filtered full genome data sets were analysed.

Motif $_{82}$KFRDI$_{86}$ was conserved in all species, as well as amino acids Glu 71 and Cys 147 of the protease catalytic triad (Fig. 3). In overwhelming majority of cases in second position of the putative RNA-binding tripeptide (position 155) was Gly.
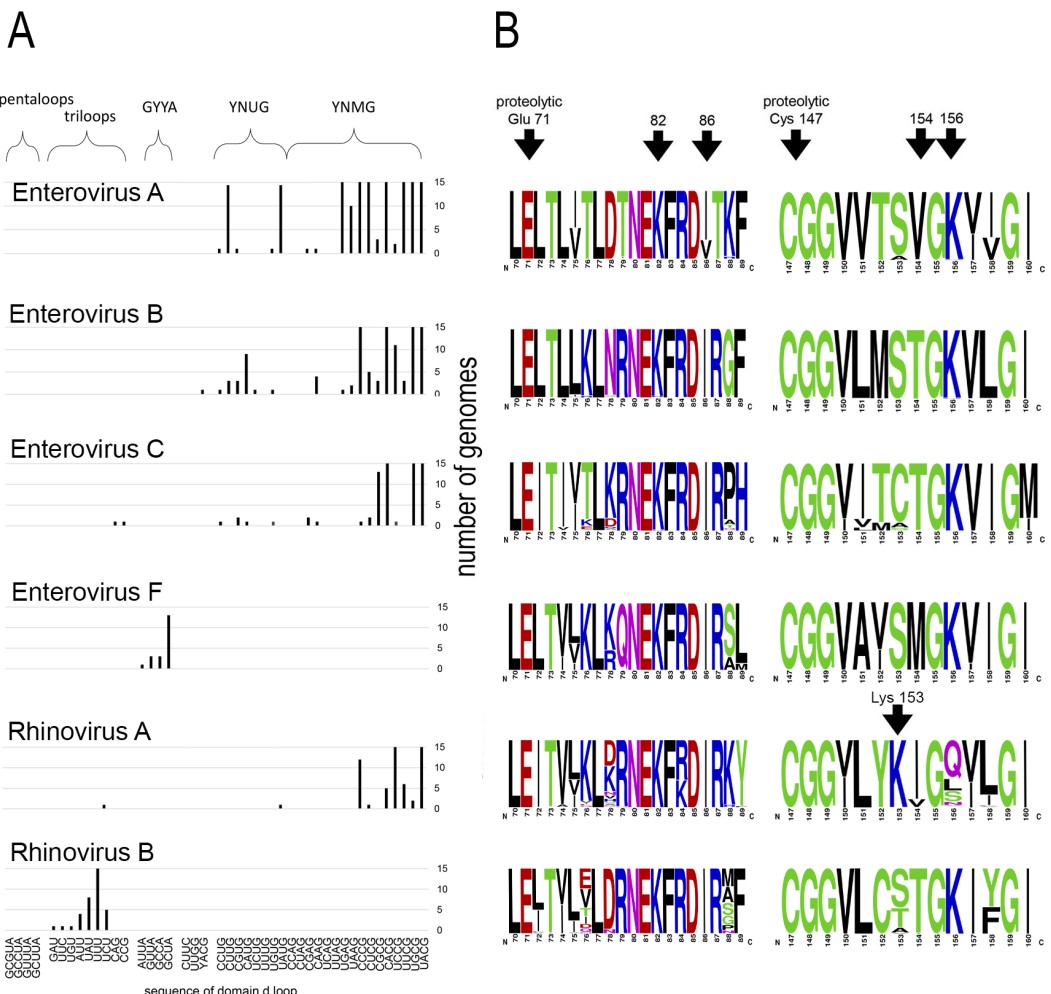

**Figure 3** **Distribituion of domain d loop sequence and amino acid motifs in the 3C protein.** (A) Distribution of domain d loop sequences. The regions corresponding to tetraloop consensuses, triloops and pentaloops are shown. Number of genomes cut off at 15 for clear view of sequence distribution. (B) The frequency plot of amino acid sequence of 3C in species of genus *Enterovirus*. The amino acid sequence logo was done with the WebLogo server (*Crooks et al., 2004*). Arrows indicate amino acids of the proteolytic triad (Glu71 and Cys 147), the first and the last amino acids of motif $_{82}$KFRDI$_{86}$, the putative RNA-binding tripeptide 154–156 of 3C and Lys153 in the protein 3C of *Rhinovirus A*.

No mutual dependence between loop sequences and tripeptide sequences was found within enterovirus genomes of the same species. For example, *Enterovirus A* genomes contained 17 unique variants of the tetraloop sequence, whereas the predominant fraction of 3C sequences (548 out of 564) contained the conservative tripeptide VGK at positions 154–156 (Fig. 3, Table S9). It is noteworthy that, this tripeptide was not found exclusively only in genomes of the EV71 serotype, although genomes of this serotype prevailed in the data set. Other *Enterovirus A* genomes contained tripeptides VGR (seven out of 564), TGK (four out of 564), IGK (three out of 564), VGE (one out of 564) and SRK (one out of 564) (Fig. 3, Table S9). Genomes with tripeptides other than VGK contained no peculiarities of the domain d loop sequence (Table S9). This observation confirms that the specific loop

sequence is not likely to be the main subject for recognition by the RNA-binding tripeptide. Similarly, all or almost all genomes of *Enterovirus B* (242 out of 244), *Enterovirus C* (272 out of 274), *Enterovirus D* (all), *Enterovirus G* (seven out of 8), *Enterovirus H* (a total of two: genomes—one with TGK, one with TGR), *Enterovirus J* (all) and *Rhinovirus B* (36 out of 37) species contained tripeptide TGK at positions 154–156 of the 3C protein (Fig. 3, Tables S7, S9 and S10). Alternative tripeptides were TGR in two genomes of *Enterovirus B* and one genome of *Enterovirus H*; IGK in one genome of *Enterovirus C* species and in one genome of *Rhinovirus B* species; PGK in one genome of *Enterovirus C* species; and MGK in one genome of *Enterovirus G* species (Tables S7, S9 and S10).

Genomes of *Enterovirus E* and *F* species contained two oriLs with tetraloops in domain d mostly of consensus GYYA or pentaloops of consensus GHBUA, where H = A/C/U and B = U/C/G. All genomes contained tripeptide MGK at positions 154–156 of protein 3C (Fig. 3, Table S7). Interestingly, a similar loop-tripeptide pair was found in one genome of *Enterovirus G* species (strain LP 54, accession number AF363455). It contained tetraloop GUUA in domain d of its single oriL and tripeptide MGK in 3C. Unlike this unique genome, other genomes of *Enterovirus G* species contained tetraloops of YNMG consensus and tripeptide TGK in the protein 3C.

Rhinovirus genomes contained tetraloops, mostly of consensus YNMG (*Rhinovirus A* and *C* species) or triloops (*Rhinovirus A* and *B*) (Table 2). *Rhinovirus A* genomes with tetraloops in domain d contained tripeptides in 3C with positively charged amino acid before the tripeptide, but not in its final position, as in case of genomes of *Enterovirus A–C* species (Fig. 3, Tables S11 and S12). The sequence of tripeptides, which did not depend on the tetraloop sequence, was, in descending order, IGQ (the most abundant, 65 genomes out of 119), IGL (20 genomes out of 119), IGS, VGS, IGN, VGQ, IGV and VGH (Table S11). In the case of the *Rhinovirus A* genome with triloop UCU in domain d (isolate V38_URT-6.3m, accession number JF285329), protein 3C contained tripeptide TGK without positively-charged amino acid before it (Table S11). All genomes of *Rhinovirus B* species contained triloops in domain d, with all but one (with IGK) containing tripeptide TGK in 3C. Genomes of Rhinovirus C contained tetraloops mostly of consensus YHCG (H = all but G) and tripeptides in 3C without a positively charged amino acid at the last position (TGN, VGN, TGH) or outside of the tripeptide (Table S12). One genome contained tetraloop CUUC paired with most abundant tripeptide, that is, TGN (23 out of 37 genomes) (Table S12).

Thus, dependence between apical domain d sequences and tripeptides in protein 3C within a species was not detected (Fig. 3). We can state that the tripeptide and motif KFRDI are almost non-variable within a species compared to the domain d loop sequence, but there is a specifically preferred tripeptide sequence for each species. Hence, tripeptide sequences are species-specific, while the domain d loop sequences are almost universal among *Enterovirus A, B, C* and *D* and *Rhinovirus A* and *C* species.

## DISCUSSION

Most of the domain d apical loops in enterovirus genomes were represented by tetraloops. The most common variants of tetraloop sequences corresponded to consensuses YNHG

(Y = C/U, N = any, H = A/C/U) (Table 2). Similar results were obtained in our previous experimental work, where eight apical nucleotides of domain d of the poliovirus genome were randomized and viable variants were selected *in vitro,* with the majority of selected tetraloops belonging to consensus YNHG (*Prostova et al., 2015*). Some tetraloops of consensus YNHG were found in genomes in the NCBI database, but not among the variants selected *in vitro*, namely tetraloops CACG, CUCG, UAAG, UGAG, CAAG, CGAG, UGUG and CUUG (*Prostova et al., 2015*). Tetraloops UGAG, UGUG and CUUG were reconstructed with a U****G flanking base pair in the context of the poliovirus genome strain Mahoney, which supported effective virus replication (*Prostova et al., 2015*).

Conversely, tetraloops UUAG, UCAG and CCAG, found in domain d of selected *in vitro* viable poliovirus variants, were able to support virus reproduction; however, they were not found in naturally circulating viruses (*Prostova et al., 2015*). One tetraloop of the YNHG consensus (CUAG) was neither found in genomes from the NCBI database ($n = 2,842$), nor in the randomized poliovirus genomes selected *in vitro* ($n = 62$) (Table 2). Thermodynamic stability is unlikely to be the reason why this and other tetraloops were unrepresented as the melting temperature of stem loops with avoided tetraloops is within range of the melting temperature of YNHG tetraloops, which supported replication (*Proctor et al., 2002*). Moreover, tetraloops UUAG and UCAG are common in rRNA (*Woese et al., 1990*). Sample insufficiency cannot be excluded for both database and *in vitro* selected sets of genomes, but it is safe to conclude that these tetraloop variants are at least extremely rare. In any case, the fact that the incidence of these tetraloops is much less than for other tetraloops indicates that such variants are possibly less fit.

The most abundant tetraloops in the domain apical region of genomes from the NCBI database and variants selected *in vitro* could be compiled into consensuses UNCG and CNCG (Table 2, Table S1). At the same time, these tetraloops are most abundant in rRNA, and, with certain closing base pairs, among the most thermodynamically stable tetraloops (*Woese et al., 1990*; *Proctor et al., 2002*). Tetraloops of these consensuses and some other found tetraloops of the YNHG consensus form a specific spatial structure of the UNCG structural class of stable tetraloops (*Cheong, Varani & Tinoco, 1990*; *Varani, Cheong & Tinoco, 1991*; *Du et al., 2003*; *Du et al., 2004*).

Another set of tetraloops, which correspond to GNYA consensus, was found both in genomes of *Enterovirus E* and *F* and in genomes of viable polioviruses selected *in vitro* (*Prostova et al., 2015*). Tetraloop GCUA was able to support the effective replication of poliovirus and, together with tetraloop GUUA, is known to assume an UNCG fold (*Ihle et al., 2005*; *Melchers et al., 2006*; *Prostova et al., 2015*). In sum, these data suggest that the spatial structure, rather than the exact sequence, is the main subject for recognition by virus protein 3C. Structure-based recognition of tetraloops occurs in several known RNA-protein complexes. For example, tetraloops with a GNRA class structure in the context of bacteriophages P22 and λ genome transcription antitermination element boxB are specifically recognized by the bacteriophage N-protein arginine-rich motif (ARM) (*Cai et al., 1998*; *Legault et al., 1998*; *Schärpf et al., 2000*). Arginines and lysines of the ARM recognize the shape of the negatively charged phosphodiester backbone of the stem-loop and positions N-peptide for hydrophobic or stacking interaction with a non-conserved

nucleotide of the loop (*Cai et al., 1998*; *Legault et al., 1998*; *Schärpf et al., 2000*; *Thapar, Denmon & Nikonowicz, 2013*). Another example of structure-specific recognition is the complex of the double-stranded RNA-binding domain (dsRBD) of RNase Rnt1p and AGNN class tetraloop (*Chanfreau, Buckle & Jacquier, 2000*; *Lebars et al., 2001*; *Wu et al., 2001*; *Wu et al., 2004*; *Wang et al., 2011*; *Thapar, Denmon & Nikonowicz, 2013*). Motif dsRBD recognizes the phosphodiester backbone at the 3′ side of the tetraloop and its non-conserved third and fourth nucleotides (*Wu et al., 2004*; *Wang et al., 2011*; *Thapar, Denmon & Nikonowicz, 2013*).

The sequence to structure degeneracy (different RNA sequences are able to form similar spatial structure) is the known phenomenon (*Petrov, Zirbel & Leontis, 2013*; *Bottaro & Lindorff-Larsen, 2017*). Moreover, it is suggested to refrain from associating sequences with a particular fold (*D'Ascenzo et al., 2016*; *D'Ascenzo et al., 2017*). Together with the literature data, our result let us assume that sequence-structure degeneracy is a universal way in which RNA tetraloops are used in nature (*Lebars et al., 2001*; *Wu et al., 2004*; *Ihle et al., 2005*; *Petrov, Zirbel & Leontis, 2013*; *D'Ascenzo et al., 2016*; *D'Ascenzo et al., 2017*; *Bottaro & Lindorff-Larsen, 2017*).

It can be speculated that pentaloops in domain d of the *Enterovirus E* genome and triloops of domain d of rhinoviruses have the potential to comprise the same UNCG fold as some YNHG and GNYA tetraloops. For HRV14 domain d, it was shown that its triloop resembles the structure of the first and last two nucleotides of UNCG structural class tetraloops (*Headey et al., 2007*). There are pentaloops with four nucleotides that belong to consensus UNCG, GNRA or gCUUGc, which are able to form spatial structures of corresponding structural classes with the fifth bulged nucleotide (*Cai et al., 1998*; *Schärpf et al., 2000*; *Theimer, Finger & Feigon, 2003*; *Oberstrass et al., 2006*; *Liu et al., 2009*). It is possible that four nucleotides of the pentaloops in domain d of Enterovirus E species have a UNCG fold with one bulged nucleotide.

Tetraloops that did not belong to the YNHG or GNYA consensus were found in both sets of natural and *in vitro* selected genomes. However, in an experiment such variants were found to evolve towards the YNHG or GNYA consensus (*Prostova et al., 2015*). Apparently, tetraloops that do not belong to the YNHG or GNYA consensus are less fit in most settings and under experimental conditions. However, as these variants may still be found in a few naturally circulating viruses (consequently, they have emerged and been fixed), we speculate that they may be beneficial under specific replication conditions.

A similar structure of domain d and its apical region suggests the free exchange of this region between genomes of the same and different species of *Enterovirus* genera. Indeed, viable intra and inter species recombinants for this region could be obtained *in vitro* (*Muslin et al., 2015*; *Bessaud et al., 2016*). To evaluate the relative impact of the high mutation rate and recombination on domain d apical loop variability, sequences of EV71 C4 genotype viruses were analysed. The natural recombination in EV71 genotype C4 is much less frequent than other *Enterovirus A* types (*Lukashev et al., 2014*); meanwhile, only one recombinant genome (accession number HQ423143) was detected in our data set. Therefore, the variability of its domain d loop sequence reflects changes that were only accumulated via mutations. The diversity of the domain d loop sequence of EV-71 C4

viruses was far less prominent than among *Enterovirus A* genomes and represented only by five tetraloop sequence variants (Table 2). As the most recent common ancestor of EV71 genotype C4 dates back about 20 years (*McWilliam Leitch et al., 2012*), this diversity, although limited, has only emerged very recently. On the other hand, the high sequence variability of the domain d apical region in all enterovirus genomes was possibly assisted by inter- and intra-species recombination events.

Interestingly, in contrast to the similar structure of domain d and the very similar distribution of its apical sequences in genomes of different enterovirus species, its putative RNA-recognition tripeptide of 3C is diverse (Fig. 3). Most *Enterovirus A* genomes contain tripeptide VGK in 3C, while there is a prevalence of the TGK tripeptide among genomes of *Enterovirus B, C* and *D* species (Fig. 3). Genomes of *Rhinovirus A* and *C* also contain common enterovirus tetraloops in the domain d apical region, but, in 3C, unlike other species, they contain tripeptides without positively charged amino acids (Fig. 3, Tables S11 and S12). Positively charged amino acids are often involved in the interaction with RNA, in particular, with phosphates of the RNA backbone. As such, they are of importance to RNA-protein recognition (*Jones et al., 2001*; *Bahadur, Zacharias & Janin, 2008*). In *Rhinovirus A* genomes, positively charged amino acid "jumped" from the last position of the tripeptide (position 156) to the position that precedes the tripeptide (position 153) (Fig. 3, shown by an arrow). The residue at position 153 starts and the residue at position 156 ends the reverse turn between beta strands dII and eII of protein 3C (*Mosimann et al., 1997*; *Matthews et al., 1999*; *Cui et al., 2011*). In a crystal structure of the Rhinovirus A2 protein 3C, the side chain of Lys153 (preceding the tripeptide) is positioned in a region similar to that of the side chain of Lys156 (in last position of the tripeptide) in the crystal structure of Enterovirus 71 and Poliovirus 1 proteins 3C (*Mosimann et al., 1997*; *Matthews et al., 1999*; *Cui et al., 2011*). Thus, Lys at position 153 of 3C has almost the same potential to interact with the RNA-ligand as Lys at position 156 (*Mosimann et al., 1997*; *Matthews et al., 1999*; *Cui et al., 2011*). Genomes of *Rhinovirus C* species do not contain a positively charged amino acid, either inside the tripeptide of the 3C protein, or in the neighbouring positions, possibly indicating that tripeptide 154–156 in the protein 3C of *Rhinovirus C* genome does not interact directly with RNA. Thus, 3C is able to recognize domain d of the oriL with tripeptides of a different sequence. In contrast to the domain d structure and its apical sequence, the tripeptide is species-specific. The diversity of the tripeptide, which is expected to recognize domain d, has several compatible explanations. Residue 154 of the tripeptide possibly does not interact with domain d directly. The tripeptide may be involved into a species-specific cooperative amino acid network (amino acid "epistasis"). Moreover, different tripeptides could reflect slightly different molecular mechanisms for domain d recognition.

The complexity of the tripeptide's role in domain d recognition can be shown in several examples. The 3C protein of different species with the same RNA-binding tripeptide is not guaranteed to bind the same structured domain d. Genomes of the *Rhinovirus B* contain triloops in the apical region of domain d, which are paired with tripeptide TGK in 3C, common for genomes with tetraloops. In contrast, protein 3C of the Coxsackie virus B3 (*Enterovirus B* species, containing tripeptide TGK) cannot recognize oriL sufficiently

well when domain d is capped with a triloop (*Zell et al., 2002*). This indicates that the sequence of the RNA-binding tripeptide is probably not the exclusive participant in oriL-3C recognition. In other words, different molecular mechanisms of oriL-3C recognition have evolved in every enterovirus species independently. For example, it was shown for Rhinovirus 14 (*Rhinovirus B* species) that protein 3C recognizes the stem region of domain d, rather than its apical loop (*Leong et al., 1993*). Another oriL-3C recognition mechanism is seemingly employed by *Enterovirus E* and *F* species, two oriLs of which play the same role as the single oriL in genomes of other enteroviruses (*Pilipenko, Blinov & Agol, 1990*; *Zell et al., 1999*). The apical loop of their domain d is a tetra- or pentaloop with a sequence that differs from the loop consensuses of other enteroviruses. The RNA-binding tripeptide in 3C is species-specific as well, and is always MGK (Table S6). Interestingly, one genome of *Enterovirus G* species had the same pair domain d loop: tripeptide of 3C, i.e., GUUA MGK. Domain d of *Enterovirus G* species is prolonged in comparison to the length of domain d in genomes of other species (*Krumbholz et al., 2002*) (Fig. 2). Tripeptide MGK in the 3C of *Enterovirus E, F* and *G* possibly indicates another molecular mechanism of oriL-3C recognition (*Krumbholz et al., 2002*). Therefore, we assume that, though putative RNA-binding, the tripeptide, in most cases, possibly interacts with the domain d apical region (since amino acid substitutions in it are known to compensate for structural disturbance in domain d); however, this interaction is not the only one that determines the evolution of oriL-3C interaction. Altogether, the data suggest that the independent evolution of the putative RNA-binding tripeptide of 3C and domain d of oriL occurs.

## CONCLUSIONS

We analysed the variety and occurrence of the replication element oriL's functional loop and its protein ligand virus protease 3C. RNA-binding motifs of 3C are species-specific, in contrast to domain d loop sequences: the sequence variety of domain d loop is almost the same for *Enterovirus A, B, C* and *D* and *Rhinovirus A* and *C* species, whereas tripeptide sequence variety differs. The conservation of the tripeptide sequence within species, together with the almost universal diversity of tetraloop sequences among species, indicates the occurrence of the independent evolution of these two elements. Our results suggest the structure-based, rather than sequence-based, recognition of domain d by virus protein 3CD. These, together with the data reported in the literature, let us assume that the sequence-structure degeneracy is a universal way in which RNA tetraloops are used in nature.

### Funding

This work was funded by the Russian Science Foundation (No. 15-15-00147). The funders had no role in study design, data collection and analysis, decision to publish, or preparation of the manuscript.

## Grant Disclosures

The following grant information was disclosed by the authors:
Russian Science Foundation: 15-15-00147.

## Competing Interests

The authors declare there are no competing interests.

## Author Contributions

- Maria A. Prostova conceived and designed the experiments, performed the experiments, analyzed the data, wrote the paper, prepared figures and/or tables, reviewed drafts of the paper.
- Andrei A. Deviatkin conceived and designed the experiments, performed the experiments, analyzed the data, contributed reagents/materials/analysis tools, wrote the paper, prepared figures and/or tables, reviewed drafts of the paper.
- Irina O. Tcelykh performed the experiments, analyzed the data, contributed reagents/materials/analysis tools, reviewed drafts of the paper.
- Alexander N. Lukashev conceived and designed the experiments, analyzed the data, contributed reagents/materials/analysis tools, wrote the paper, reviewed drafts of the paper.
- Anatoly P. Gmyl conceived and designed the experiments, wrote the paper, reviewed drafts of the paper.

## Data Availability

Multiple alignment files (.fas), containing Enterovirus full genomes before and after filtration, are provided as Supplementary Files.

## Supplemental Information

Supplemental information for this article can be found online at http://dx.doi.org/10.7717/peerj.3896#supplemental-information.

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
