# Peer review of "Independent evolution of tetraloop in enterovirus oriL replicative element and its putative binding partners in virus protein 3C"

_PeerJ, doi:10.7717/peerj.3896_

## Round 0.1 · original submission · Minor Revisions

Your manuscript has been reviewed by three experts and the overall comments are very positive. However, there are some minor issues that need to be addressed before acceptance. I am happy to receive a new version of the manuscript with all the corrections suggested by the referees. I am sure their input will greatly contribute to make your paper even better.

Thank you

·

Basic reporting

The language used by Prostova et al. in their paper is clear and unambiguous, yet I would like to recommend a further checking to improve wording/phrasing in some sections to improve the overall readability. Some specific notes on this regard:
Line 43: I suggest adding long at the end of the sentence “… with plus strand genome about 7500 nt long”
Line 60: I suggest changing "leafs" to "leaves"
Line 69: I suggest changing "extremal" to "extreme"
Line 107: I suggest changing "formed" to "created"
Lines 119-122: I suggest expressing the total sequences per organism in the clearer format organism (N total)
Line 144: I suggest changing "conservative" to "conserved"
Line 299: please remove the comma
Line 398: I would say that data "suggests" rather than "demonstrates"

Overall, the submission is ‘self-contained’ with a clear aim and clear results, which can be used for future experimental validation of the hypothesis made by the authors in the paper.
The authors made a good job in properly referencing and giving relevant background to their work. Their paper conforms to the journal guidelines, figures are clear and the raw data is appropriately given as supplementary material. To improve their manuscript, I would like to suggest to the authors to expand the Conclusions section to include more details on the final message of their paper.

Experimental design

The work successfully qualifies for the Aims and Scope of the journal, conveying new findings on a relevant biological system using widely known bioinformatics techniques. The research question is well defined in the first sections of the paper, with several mentions of the significance of their study from various biological points of view. However, I wish the authors would put more emphasis on how their study contributes to fill the knowledge gap they are trying to address.
The methodology used and described in the paper is sound enough to support their conclusions, though I would suggest to the authors to give more details on some methodological approaches that can be relevant to make their work reproducible. For instance in line 94 the authors report they used Clustal for sequence alignments, but without giving more details on the parameters that have been used (and if they differ between RNA and protein analysis). Another example is in lines 113-114 where the Vienna RNA Websuit has been used for secondary structure fold. I suggest adding some more details about the algorithm and parameters that have been used.

Validity of the findings

The data presented by the authors are clear and robust, which support their conclusions. The speculative part is somewhat limited and identified as such, without falling in a common pitfall in this type of study of over interpretation of raw bioinformatics data to extract stretched biological meanings. A note I wish to make to the authors is that starting from line 312 in the Discussion they highlight how their data suggest a recognition structure-based rather than sequence-based, which to me is worth to be mentioned again towards the Conclusions part, being a significant result for general macromolecular recognition rules.

Additional comments

All things considered, the work by Prostova et al. is in my opinion suitable to be published in PeerJ and can be further improved with some minor revisions/improvements I mentioned in the previous sections

·

Basic reporting

Overall the paper is clear and there are no widespread severe language issues present. However, there are some small grammatical and wording issues. I have listed a few example below.

1. On line 69 the phrase 'extermal stability' should be 'extreme stability'.

2. On line 215: 'the conservative motif KFRDI' should be 'the conserved motif KFRDI'.

3. On line 228: 'tripeptide (position 155) was invariantly Gly.' should be 'tripeptide (position 155) was always Gly.'

This is not a comprehensive list of all such issues. Overall the paper could use some small editing for these minor wording issues.

The introduction and background provide a clear summary of the subject and enough context to understand the question being asked. In addition, the paper is well structured and organized. The paper as a whole is an easy read.

In addition the figures are generally good. Figure 1 is very helpful for understanding the biological context of the research and Figure 2 helps in understanding the finer details of the loop in question. However, there are some issues with Figure 3. They are:

1. Part A of the figure may best be presented as a simple table. This would give more complete information to the reader as it would be easier to see the exact counts of type of loop.

2. Part B needs some modifications. The arrows in B that indicate the important positions in the protein are very helpful it would be ideal if the number in the sequence logo was consistent with the stated numbers. That is the position labeled with 'proteolytic Glu 71' should be number 71 instead of 2. Finally, it would be helpful to label the arrow in the lower right as Lys153 from Rhinovirus A as it is not clear that that motif is from Rhiovirus A.

Overall, the paper is well presented but some edits are necessary.

Experimental design

The authors aim is to find a correlation between sequence conservation of domain d loop regions in Enteroviruses and the interacting amino acid sequences from protein 3C. They used a simple straightforward method to find any correlation. The methods section explains much of the techniques used. However there are some issues with their methods:

1. They do not justify the inclusion of otherwise excluded loop sequences in their analysis. For example, on line 171 the authors mention adding 3 unique tetra loop sequences back into the data set after filtration. I suspect this is to maintain diversity of loop sequences from the species. However, there is it is not clear to me why this must be done. The reasons for this decision must be clearly justified in the text. In addition, I would be interested to know if their conclusions change when these sequences are included.

2. They use a 99% percent cutoff and claim that a 95% cutoff results in a 'dramatic loss of unique tetraloop variants', providing a number of sequences lost would help justify the selection of 99% cutoff.

In summary, I believe their methods are correct but some details require explanation.

Validity of the findings

The authors conclusions about lack of a relationship between the sequence conservation in the domain d loop and the interacting protein are well supported by the data. As they state there is no clear pattern of loop sequence to protein sequence conservation. In addition, their observation that the protein sequence appears to be more diverse than the RNA sequence is both interesting and supported. The authors may wish to emphasize this contrast more.

However, the authors may be underestimating the degeneracy of the sequence to structure relationship. In Petrov et. al (doi:10.1261/rna.039438.113) it was found that some loops may differing sequences as well as number of nucleotides but form the same overall 3D structure. Observations such as this strength the their conclusions and the authors may wish to cite such work.

In addition, there has been some work studying the structure RNA/protein interactions. The authors may want to review this literature to suggest a how the diverse protein and loop sequence interact. If their suggestion that the overall 3D structure is maintained then the interaction should also be maintained. This may be outside the scope of the current work and I understand if the authors wish to leave such work for another publication.

Reviewer 3 ·

Basic reporting

The study is well-conducted and detailed. I have only few minor comments that might clarify the manuscript.

1) Table 2: It would be more clear if proportions of loop sequences instead (or in addition) to occurrence was indicated in the heatmap (i.e. percentage of a given loop sequence out of sequences observed in a given virus species).

In addition, it would be interesting to see if there are statistically significant differences in the proportions of a given loop sequence between virus species.
For example, loop sequence CUCG seems to be prevalent in EV-A species, but not in EV-B and EV-C species.

Table S1 contains more information than Table 2 (regarding dataset bias - i.e. predominance of EV-A71 and PV), therefore I would prefer showing Table S1 in the main article.

In Table S1, it would be more clear if two more columns; 'non-A71 EV-A' and 'non-polio EV-C' would be included. This would help the reader to understand the structure of the data (i.e. how much of the diversity in 'all' column is explained by EV-A71 and PV).

2) Fig 3: Only five species are shown - should others be included? e.g. rhinovirus B (with triloops) would be informational.

3) Supplementary Tables:

Should protein 3C tripeptide sequences be also given for EV-B, EV-D, EV-E, EV-F and Rhinovirus B?

Experimental design

no comment

Validity of the findings

1) Page 13 line 164-165:
"Interestingly, the diversity of tetraloops among EV71 serotype was similar to the diversity of tetraloops in the whole Enterovirus A species (Table S1)."

This statement needs clarification. According to Table S1 it seems that some tetraloops (such as UUCG, UAAG and UGAG) are found only in EV-A71 type but not in the other EV-A types.On the other hand, tetraloop UGCG seems to be highly prevalent among other EV-A types (N=112), but rare among EV-A71 (N=2), whereas CUCG seems to be prevalent among EV-A71 (N=127) but rare among the other types (N=5).

Additional comments

Generally, the study is well-conducted and detailed. The conclusions are supported by the data. I have only few minor comments that might clarify the manuscript.

---

## Round 0.2 · accepted · Accept

I have detected a couple of grammatical errors that you should resolve while in production.They are in lines 345-346: "...our results let us" or "... our result lets us" and in lines 445-446 ...way in which RNA tetraloops...". Thank you.